# Partial *fads2* Gene Knockout Diverts LC-PUFA Biosynthesis via an Alternative Δ8 Pathway with an Impact on the Reproduction of Female Zebrafish (*Danio rerio*)

**DOI:** 10.3390/genes13040700

**Published:** 2022-04-15

**Authors:** Zuzana Bláhová, Roman Franěk, Marek Let, Martin Bláha, Martin Pšenička, Jan Mráz

**Affiliations:** South Bohemian Research Center of Aquaculture and Biodiversity of Hydrocenoses, Faculty of Fisheries and Protection of Waters, University of South Bohemia in České Budějovice, Zátiší 728/II, 389 25 Vodňany, Czech Republic; zblahova@frov.jcu.cz (Z.B.); franek@frov.jcu.cz (R.F.); mlet@frov.jcu.cz (M.L.); blaha@frov.jcu.cz (M.B.); jmraz@frov.jcu.cz (J.M.)

**Keywords:** *fads2*, Δ6/Δ5/Δ8 desaturase, long-chain polyunsaturated (LC-PUFAs) biosynthesis, CRISPR/Cas9, zebrafish (*Danio rerio*) reproduction, dihomo γ-linolenic acid (DGLA)

## Abstract

The zebrafish (*Danio rerio*) genome contains a single gene *fads2* encoding a desaturase (FADS2) with both Δ6 and Δ5 activities, the key player in the endogenous biosynthesis of long-chain polyunsaturated fatty acids (LC-PUFAs), which serve essential functions as membrane components, sources of energy and signaling molecules. LC-PUFAs include the precursors of eicosanoids and are thus predicted to be indispensable molecules for reproductive health in virtually all vertebrates. In mice, an amniotic vertebrate, *fads2* deletion mutants, both males and females, have been confirmed to be sterile. In anamniotic vertebrates, such as fish, there is still no information available on the reproductive (in)ability of *fads2* mutants, although zebrafish have become an increasingly important model of lipid metabolism, including some aspects of the generation of germ cells and early embryonic development. In the present study, we apply the CRISPR/Cas9 genome editing system to induce mutations in the zebrafish genome and create crispants displaying a degree of *fads2* gene editing within the range of 50–80%. Focusing on adult G_0_ crispant females, we investigated the LC-PUFA profiles of eggs. Our data suggest an impaired pathway of the LC-PUFA biosynthesis of the ω6 and ω3 series in the first-rate limiting steps of the conversion of linoleic acid (LA) into γ-linolenic acid (GLA), and α-linolenic acid (ALA) into stearidonic acid (SDA), respectively, finally resulting in bad-quality eggs. Our data suggest the existence of an alternative Δ8 pathway, which bypasses the first endogenous LC-PUFA biosynthetic step in zebrafish in vivo, and suggest that the zebrafish bifunctional FADS2 enzyme is actually a trifunctional Δ6/Δ5/Δ8 desaturase.

## 1. Introduction

Long-chain polyunsaturated fatty acids (LC-PUFAs) are of high significance in healthy state maintenance and the reproduction success of virtually all vertebrates. Being important building blocks of lipids and playing vital roles as integral components of cell membranes, LC-PUFAs affect membrane fluidity, regulate ion channels, modulate endocytosis and exocytosis as well as hormonal activity, have immunological effects and influence gene expression during development. As precursors of eicosanoids, some LC-PUFAs are necessary for spermatogenesis and folliculogenesis [1,2,3,4]. In oocytes, the lipid metabolism is critical for their growth, fertilization, and the development of early embryos. The improper delivery of lipids to oocytes leads to abnormal oocytes, a low egg production rate, and low viability of embryos [5,6,7].

Vertebrates are usually capable of biosynthesizing LC-PUFAs to extend endogenously from essential precursors, in addition to the main acquisition via diet [8,9]. In fast developing tissues in the early stages of life, the demand on LC-PUFAs is especially high; correspondingly, the expression from genes encoding desaturases and some other genes of the LC-PUFA biosynthetic pathway is high [10]. The precursors linoleic acid (LA) and α-linolenic acid (ALA) of ω6 and ω3 series LC-PUFAs, respectively, are essential. Products of their downstream conversions are generally considered as semi-essential. The endogenous pathway is considered to fulfil the function of a compensation apparatus under nutritionally fluctuating conditions.

The consensus LC-PUFA endogenous biosynthetic pathway involves alternating steps of desaturation (the introduction of an additional double bond inside the substrate poly-carbon fatty acid chain) and elongation (the addition of two carbons), whereby in particular steps of the biosynthesis of the ω6 and ω3 series, substrates compete for the same enzymes. In the first and rate-limiting step, FADS2 converts C18:2 Δ9,12 (LA) and the C18:3 Δ9,12,15 (ALA) into C18:3 Δ6,9,12 (γ-linoleic acid, GLA) and C18:4 Δ6,9,12,15 (stearidonic acid, SDA), respectively. Downstream enzymatic steps in the biosynthesis are shown in Figure 1 [11]. Alternative Δ8 pathways are depicted as well, however, they remain less understood. The Δ8 pathway includes the initial elongation of essential precursors requiring the action of elongase (Elovl) enzymes. Elovl5 enzymes isolated from many fish species have been shown to possess elongation activity towards LA and ALA [12,13,14,15]. The elongation products C20:2 Δ11,14 (eicosadienoic acid, EDA) in ω6 LC-PUFA biosynthesis and C20:3 Δ11,14,17 (eicosatrienoic acid, ERA) in ω3 LC-PUFA biosynthesis undergo further desaturation by FADS2 enzyme acting, in this case, with Δ8 specificity [11,16].

The developmental origins of the health and disease paradigm propose that, in the early stages of life, nutrition plays a critical role in establishing later-life health and susceptibility to diseases, as well as in some developmental disorders. It is crucial for embryonic success and for the long-term health of the organism that the yolk lipid content and utilization is not compromised [17]. Nonetheless, little is known about the influence of fatty acid composition on vertebrate embryogenesis and development. Recently, scientific reports transpired, which confirm the presence of active lipid biosynthetic processes in the first stages of life in which, among others, LC-PUFAs arise [18]; however, the significance of the endogenous LC-PUFA biosynthesis inside a yolk cell appreciates elucidation.

In humans, numerous observational studies have shown a link between childhood developmental disorders and LC-PUFA intake and ω6: ω3 LC-PUFA series imbalances, such as some neurocognitive disorders or vision development [4,10,17]. Alterations in the cellular content of fatty acids can lead to the modulation of cellular signaling and eicosanoid production [19]. For ethical and practical reasons, however, the metabolic processes cannot be studied in the human embryo. Rodents, mainly mice, are typical model systems of a primary choice, which might not be the case in investigations regarding lipid metabolism for the following reasons.

Although rodents and humans are similar in many aspects of developmental biology and physiology, there are substantial differences between them in lipid metabolism [20,21]. Rodent embryos lack maternally supplied yolk as they survive in an environment of dynamic nutrition by being completely surrounded by metabolically active inverted yolk sac epitheliums. Hence, they do not totally rely on the bulk of nutrition provided by the mother. In contrast, human embryos have protruding yolk sacs that are more structurally similar to those occurring in some teleost fish, such as the popular model zebrafish. Their yolk sac contains the finite bulk of nutrients with which they sustain metabolic functions and growth, until the onset of placental–fetal exchange in humans or exogenous feeding in fish [21].

In addition, zebrafish exhibit features that have allowed them to gain steadily growing popularity in the lipid research community to use be used as the favorite model organism. Some of their popular features are the large number of offspring (one female zebrafish produces clutches containing 100 eggs on average), allowing for relatively robust sampling with synchronized external fertilization [22], followed by rapid development, and the simplicity and efficiency of genetic manipulations, such as the modern clustered regularly interspaced short palindromic repeat (CRISPR)-associated (CRISPR/Cas) system for genome editing [23,24]. Another feature is the optical transparency of the developing zebrafish, of which the relatively newly developed imaging techniques have taken advantage, enabling the direct visualization of lipids and uncovering the to date unseen lipid trafficking routes during the very early stages of life [18]. An insight into one single individual intact fish embryo has recently become possible by ambient ionization techniques, such as desorption electrospray ionization mass spectrometry and nano-electrospray mass spectrometry, enabling the direct analysis of lipids in individual zebrafish embryos [25]. Evidently, during the last few years, the insight into zebrafish lipid metabolism has significantly increased and become a powerful tool in lipid research, with implications of more diverse taxa, including humans.

Substrate specificities of FADS2 differ among fish species, and monofunctional and bifunctional desaturases with Δ6, Δ5, and Δ4 activities have been described [26]. The FADS2 capability of Δ8 activity has been described for both mammals [16] and fish [27]. Hence, two pathways are suggested to biosynthesize LC-PUFA EPA and ARA: the “classical” Δ6 pathway (Δ6 desatutation → elongation → Δ5 desaturation) and an alternative Δ8 pathway (elongation → Δ8 desaturation → Δ5 desaturation). In zebrafish, FADS2 represents the sole enzyme with desaturation activity towards PUFAs [28]. Using reverse genetics approaches, such as modern CRISPR/Cas9 technology, the zebrafish *fads2* gene represents the single target for manipulating desaturations.

Surprisingly, there is currently no study that studies fish *fads2* mutants in oocyte development. There is a vast amount of information available on the dietary lipid manipulation of brood stock allowing developing strategies for improving spawning performance and egg quality [29]. However, it is not clear if the role of the endogenous production of LA and ALA conversion downstream products is dispensable under the nutritionally rich conditions, hence replaceable by LC-PUFA biomolecules delivered by the diet, or the well-functioning endogenous LC-PUFA biosynthesis represents a prerequisite for reproduction health. Recently, a study appeared that confirmed the maternal transfer of *fads2* mRNA to the developing oocyte in gilthead seabream (*Sparus aurata*) [30]. We deduce that the eggs’ fatty acid profile putative abnormalities are solely caused by the mother organism in fish. Using CRISPR/Cas9 technology, we determined the start codon of the *fads2* gene in zebrafish one-to-four-cell stage embryos, raised them until adulthood and determined the degree of derived crispant mosaicisms. Female crispants displaying a degree of *fads2* gene editing higher than 50% were spawned. The eggs spawned by crispant females proceeded to lipid extraction, and the fatty acyl methyl ester preparations and LC-PUFA profiles were examined with special focus on the first steps of the biosynthesis of LC-PUFAs.

## 2. Materials and Methods

### 2.1. Ethics Statement

All experiments on the animals were conducted at the Faculty of Fisheries and Protection of Waters (FFPW), University of South Bohemia in České Budějovice, Vodňany, Czech Republic. The facility has the competence to perform experiments on animals (Act no. 246/1992 Coll., ref. number 16OZ19179/2016-17214) in accordance with the principles based on the European Union Harmonized Animal Welfare Act of the Czech Republic.

### 2.2. sgRNA Design and sgRNA/CRISPR/Cas9 Complex Preparation

Our aim was to introduce mutations directly into the ATG start codon of the *fads2* gene to avoid the transcription of the FADS2 protein from the beginning. To achieve this, we used the CRISPR/Cas9 genome editing system consisting of endonuclease Cas9, which unwinds the genomic DNA duplex and creates double-strand breaks, and the single guide sgRNA, which is responsible for the site-specific recognition of the intended cleavage site. Prior to this, the synthetic sgRNA 5′-GTTCAGAGATCAGCGATGGG-3′ was designed according to the zebrafish *fads2* gene (zfin.org/ZDB-GENE-011212-1, accessed on 1 April 2022) and purchased from the SYNTHEGO company (Redwood City, CA, USA) (synthego.com, accessed on 1 April 2022); we made sure that sequences flanking the ATG start codon of our maintained zebrafish matched the sequence published on zfin.org. We PCR amplified the part of *fads2* gene using our PCR primers Fw 5′-TTTCCACCACGATCCACTGA-3′and Rev 5′-CAGTGGGTGGTGGTGGAG-3′, then gel-purified and sequenced (Macrogen) the amplicon. The Cas9 nuclease protein was freshly prepared before use, according to the manufacturer’s protocol (Cas9 Nuclease Protein, Applied Biological Materials Inc. (Richmond, BC, Canada), abmgood.com/cas9-nuclease-protein-k008-vin.html, accessed on 1 April 2022).

### 2.3. Fish Source and In Vitro Fertilization

AB-line zebrafish (*D. rerio*), reproductively mature, were maintained in a zebrafish housing system (ZebTec^®^—Tecniplast system, Buguggiate, Varese, Italy) at a temperature of 28 °C and photoperiod 14L:10D, and fed twice daily with a standard diet (GEMMA Micro, http://www.zebrafish.skrettingusa.com, accessed on 1 April 2022). Prior to artificial fertilization, breeding pairs were transferred into spawning chambers in the afternoon before the spawning, whereby the male was separated from the female by a translucent plastic barrier. In the morning, the fish were anesthetized in a 0.05% tricaine solution (Ethyl3-aminobenzoate methane sulfonate). The stripped milt from the male zebrafish was examined for sperm motility under a Nikon SMZ745T stereomicroscope (Nikon, Tokyo, Japan). The pooled milt was added to an Eppendorf tube with 50 µL of an immobilizing solution (Kurokura 180 solution). The fish were transferred into fresh water for recovery. For the in vitro fertilization, the egg batches composed of approximately 100 ovulated oocytes were collected from the females by hand stripping and separately distributed into Petri dishes. Immediately, 10 µL of the pooled sperm and 200 µL of tank water was added, followed by gently shaking for about 45 s. Finally, the fertilized eggs of the control and treatment groups were gently distributed into Petri dishes.

### 2.4. Microinjection of the sgRNA/CRISPR/Cas9 Complex into the In Vitro Fertilized Eggs, and Fish Maintenance

All the components and sterile instruments were prepared to enable the rapid delivery of the sgRNA/CRISPR/Cas9 complex into the just-fertilized egg, ideally at the one-cell stage to minimize the obstacle providing CRISPR/Cas9—the mosaicism in the threated organism. Practically, we continued with microinjections until the 4-cell stage. Microinjections were performed as follows: approximately 50 eggs (one half of the batch following the fate of the samples) were injected. The final mixture was injected through the chorion inside each egg cell using the glass micropipette with a tip of 10 µm prepared from a glass needle (Drummond, Tokyo, Japan) by the puller (PC-10; Narishige, Tokyo, Japan). The manipulation with the full needle was achieved by the help of a hand manipulator under the microscope and under a pressure of 0.100 kPa/s developed by the injector. The eggs were maintained at 28 °C in the laboratory incubator. The unfertilized, undeveloped eggs and eggs missing the fluorescent signal due to the absence of 1% FITC-biotic-dextran, a confirmation of the introduced microinjection needle content, were continuously harvested and water was exchanged. The injected embryos were subsequently raised until sexual maturity, whereby, in the larval stage, they were transferred to ZebTec system for the genetically modified zebrafish.

### 2.5. Genomic DNA Isolation, the Detection and Quantification of the CRISPR/Cas9 System-Mediated Gene Editing

Genomic gDNA was extracted from fin clips, according to the manufacturer’s protocol (Exgene^TM^Genomic DNA micro, GeneAll^®^, Seoul, Korea) from each fish under anesthetization in 0.62 mM tricaine. Genomic regions flanking the CRISPR target codon start site was PCR amplified using a high-fidelity DNA polymerase. The sequences of the PCR primers are Fw 5′-TTTCCACCACGATCCACTGA-3′and Rev 5′-CAGTGGGTGGTGGTGGAG-3′. PCR products were electrophoretically separated in 1% agarose gel, followed by gel extraction (E.Z.N.A.^®^ Gel extraction Kit, VWR, Radnor, PA, USA) and Sanger sequencing (Macrogen-Europe). DNA sequencing chromatograms were analyzed using the Inference of CRISPR Edits (ICE) free software tool (synthego.com/products/bioinformatics/crispr-analysis, accessed on 1 April 2022).

### 2.6. The Eggs’ LC-PUFA FAMEs Profiles and Statistical Analyses

Ovulated eggs from wild-type and crispant females were collected in two repetitions (M1, M2) and sent to the customer service laboratory (BIOCEV, Charles University in Prague) for lipid extraction and the esterification into fatty acid methyl esters (FAMEs), followed by LC-PUFA analysis by comprehensive two-dimensional gas chromatography with a mass detector (GCxGC/MS). The FAME samples were prepared according to a previously used methodology [31]. Primary column Tr-FAME, 59 m, 250 µm ID, and 0.25 µm PT (Thermo, Waltham, MA, USA) was coupled to a secondary column Rxi-5MS, 1.2 m, ID—250 µm ID, 0.25 µm PT (Restek, Stockbridge, GA, USA). The temperature program was as follows: 90 °C (1 min); 10 °C/min; 140 °C (0 min); 4 °C/min; 200 °C (0 min); 10 °C/min; and 250 °C (6 min). The other parameters were set as follows: the flow was 1.2 mL/min; injection temperature was 240 °C; transfer line temperature was 280 °C; modulation period was 3 s; offset between the primary and secondary columns was 10 °C; and the hot-pulse time was 1.1 s. The mass detector was equipped with an Electron Ionization and a Time-Of-Flight analyzer. The temperature of an ion source was set to 280 °C. The analyzer scanning range was m/z 29–600. The data files were automatically processed in ChromaTOF software v4.7. FAMEs were identified comparing a two-dimensional retention behavior and mass spectra with commercial FAME standards (GLC 744, NU-CHEK-PREP) and mass spectra. The percentage distribution of LC-PUFA FAME signals in each sample of egg batch was counted.

The data were analyzed in R-studio software (R ver. 4.1.1) using default libraries and the pgirmess library.

The significance of the differences among the treatments in individual LC-PUFA ratios were tested using the parametric one-way ANOVA or non-parametric Kruskal–Wallis test, if the homoscedasticity assumptions were not fulfilled—this was tested by the Shapiro–Wilk normality test and Bartlett test of the homogeneity of variances (α = 0.05). Multiple comparison tests were applied post hoc.

### 2.7. Phenotype of the Fertilized Eggs Produced by Crispant Females

Breeding pairs of reproductively mature wilt-type and crispant zebrafish (with a knockout score confirmed to be above 50%) were transferred from the housing system (ZebTec^®^—Tecniplast system, Buguggiate, Varese, Italy) into spawning chambers in the afternoon before the spawning fish (one male and one female) were separated from each other by a translucent plastic barrier. On the onset of light on the next day, the barrier was removed and the fish were observed for oviposition. A plastic mesh just above the bottom protected the eggs from being eaten by the parents. Following the spawning, the eggs were placed on Petri dishes and cultured at 28.5 °C in an incubator.

## 3. Results

### 3.1. sgRNA Design

The sgRNA sequence 5′-GUUCAGAGAUCAGCGAUGGG-3′ is the best hit, matching the requirements of being the early coding region of the gene targeted with minimum off targets and, at the same time, being a common exon and having a high activity, as shown in Figure 2.

### 3.2. The Detection and Quantification of the CRISPR/Cas9 System-Mediated Gene Editing

Mosaic founder mutant females of the G_0_ generation were identified by genotyping. Sanger sequencing data files from the controls and each single crispant fish were delivered. We analyzed them using the free available software Inference of CRISPR Edits (ICE) tool, and calculated the overall editing efficiency and determined the profiles of all the different types of edits that are present and their relative abundances. A representative result is shown in Figure 3.

The relative contribution of each indel, percentage of indels, and mutagenesis efficiency (knockout score) were counted for each crispant fish. Only the crispants with the a knockout score above 50% were selected for further research and raised until adulthood

### 3.3. Egg Quality, LC-PUFA FAME Profile Analysis and Statistical Analysis

Egg sampling, storage, transportation, and manipulation has been optimized to obtain strong signals of prepared particular LC-PUFA FAMEs using the comprehensive two-dimensional gas chromatography. The percentage distribution of LC-PUFA FAMEs (particular LC-PUFA FAMEs correlated to the total PUFA FAMEs extracted) was used for the downstream statistics. All non-parametric tests of the differences among the treatments in individual response variables resulted significantly (*p* < 0.05; Table 1), except for C20:5, EPA. Post hoc multiple comparisons revealed that all the ratios were significantly different (*p* < 0.05) between the controls and the M1 or M2 groups (Figure 4).

Our data demonstrated an increase of C20:2 Δ11,14 and decrease in GLA in the ω6 pathway, and an increase of C20:3 Δ11,14,17 and decrease in the SDA in ω3 pathway (Figure 5). This indicates the existence of a conversion of LA and ALA in the alternative Δ8 pathway of LC-PUFA biosynthesis in zebrafish partial *fads2* mutants. In comparison to the wild types, *fads2* mutants are significantly less able to desaturate LA and ALA in the Δ6 position.

It is worth noting that AA, EPA, and DHA, generally considered as the most important polyunsaturates produced in LC-PUFA biosynthesis, did not significantly differ in their AA/LA and EPA/ALA and DHA/ALA ratios between the crispants and wild types. AA, EPA, and DHA are abundant in the artificial diet given ad libitum to fish. Nonetheless, even if the *fads2* gene depletion was not complete in our G_0_ crispants (mutational rate: 50–80%) and their metabolism proved to maintain the production of main LC-PUFA biomolecules (bypassing the first desaturation step), the reproductive capabilities of females were influenced.

### 3.4. Altered Phenotype of the Embryos from a Fads2 Mutant Female x WT Male Inter-Crosses

The eggs fertilized successfully in the control WT female x WT male and crispant female x WT male inter-crosses were incubated in laboratory conditions at 28 °C for approximately 24 h. Embryos obtained from crispant female x WT male inter-crosses were noticeably brownish in color and more densely packed in chorion, in comparison with the control WT when the embryos from both groups were aligned on the same Petri dish next to each other (Figure 6).

## 4. Discussion

In the present study, we investigated the functional link between the delta-6 desaturase gene (*fads2*) partial knockout and impaired reproductive success of zebrafish (*D. rerio*) females, whereby we observed that the *fads2* partial zebrafish knockout diverts from the “classical” LC-PUFA biosynthesis and converts C18 essential substrates LA and ALA via an alternative Δ8 pathway instead of the Δ6 desaturation pathway.

Based on the previous studies performed on mammals, infertility has been marked as a hallmark of FADS2 deficiency. It was confirmed that *fads2* mouse females are sterile [4,32]. Mammals are amniotic organisms and display substantial differences in reproductive functions, with regard to lipid metabolism from anamniotic organisms [20,21]. Indeed, to extrapolate lipid research conclusions between these diverse groups of organisms might be questionable, until a functional study of this phenomenon is conducted.

We used the popular model organism, zebrafish, applied the CRISPR/Cas9 genome editing tool to target the start codon of the *fads2* gene, raised generated potential crispants until adulthood, and then focused on the reproductive performances. Progenies of the crispant x crispant inter-crosses were not viable in our hands, and did not succeed to proceed to biallelic mutant generation. However, previous research confirmed that even if CRISPR-targeted G_0_ generation fish were genotypically complex, the highly induced mutagenesis frequencies of somatic mutations could still result in obvious phenotypes in G_0_ founders in zebrafish [23,33,34]. Taking the observations into account, we mated crispant females with wild-type males, with the aim to investigate a putative impact on egg quality. The oocyte development fully relies on the maternal nutritional supply, which consists of a large portion of lipids for which LC-PUFAs are of high importance. Zebrafish embryos have been confirmed to obtain *fads2* mRNA from the mother [30], hence the mother’s lipid metabolism is supposed to be responsible for the expected phenotypic prevalence of the *fads2* gene function.

In our study, we included zebrafish females, in which we determined the proportion of cells that have either a frameshift or 21+ bp indel (KO score) above 50%. To analyze his, we used the ICE software, for which the effectiveness has been rigorously evaluated by the Synthego company by analyzing thousands of edits performed over multiple experiments and comparing the robustness and accuracy [35,36]. The undeniable advantage of this tool for our study was that there was no need to euthanize the fish for the gene expression analysis to quantify the mRNA levels, which, at the same time, could be considered as a weakness of this study. Evidently, our study would greatly benefit from the transcriptomic analyses of zebrafish *fads2* partial knockouts.

Previous research has provided substantial evidence on the significant effects of dietarily obtained LC-PUFA levels and balance on brood stock reproductive success. However, the effect of LC-PUFAs with an endogenous LC-PUFA biosynthesis origin has been inadequately studied. Therefore, we fed zebrafish with a standard diet (GEMMA Micro, Westbrook, MA, USA. http://www.zebrafish.skrettingusa.com, accessed on 1 April 2022), with the aim to observe the phenotypes caused by impairments in the endogenous LC-PUFA biosynthesis, for which the fish itself cannot compensate, even if the diet is provided ad libitum. Endogenous production has been mainly hypothesized to serve as a compensation apparatus, which helps the organisms to maintain homeostasis under fluctuating environmental conditions and LC-PUFA availability. However, we hypothesized that endogenous biosynthesis performs functions that are not yet fully understood, and that our zebrafish *fads2* partial knockout model with a KO above 50% offers a unique insight into the organism that faces the situation of desaturation products not being completely depleted, yet significantly (without the depletion of essential LA and ALA at the same time) encourages the metabolism to undergo compensational measures, which are not known.

One important prerequisite for the present study is the absence of homologous genes encoded in zebrafish genomes, as well as the absence of functional splicing variants, both of these being investigated in the previous research. In zebrafish, only a single *fads* gene has been identified, which has been demonstrated to exhibit activity similar to that of both mammalian delta-6 and delta-5 desaturases [28,36]. Moreover, the zebrafish genome does not code a homologous gene to *fads3*, which could compensate for the Δ6 and/or Δ8 desaturation loss of function [37]. In *fads2*-null mice (−/−), no in vivo AA synthesis was detected after the administration of [U^13^C] linoleic acid (LA), indicating the absence of the Δ6 desaturase isozyme [38]. Their study also showed the absence of the Δ6 desaturase isozyme for the first and last steps of desaturation in DHA and DPA ω6 LC-PUFA biosynthesis. Based on the current knowledge, we do not consider any Δ6 desaturation activity present in zebrafish performed with another enzyme. Theoretically, it should be considered that the targeting of the *fads2* gene start codon could cause the destruction of only one splicing variant by maintaining the possibility of the transcription of another splicing variant from an alternative start codon, which could be partly responsible for the presence of the activity of Δ8 desaturation. We do not consider that this occurred, based on the data delivered by Sibbons et al. [39] who excluded the alternative splicing of *fads2* gene.

According to our results, the fact that zebrafish face the situation of a significantly lower abundance of functional *fads2* genes makes decisions to keep the gene product to act only in those reactions, which are the most important for the maintenance of a healthy state. We observed that the mutants preferentially convert essential substrate LA and ALA via an alternative Δ8 pathway of LC-PUFA biosynthesis, bypassing the initial Δ6 desaturation step in which GLA and SDA are produced. It is probable that following the alternative Δ8 pathway saves more Δ6 desaturase enzyme molecules and maintains a higher capacity for downstream polyunsaturated products of LC-PUFA biosynthesis, such are the AA, EPA, and DHA. Hence, as a result of the improper functioning *fads2* gene, which cannot provide a sufficient amount of appropriate desaturation reactions, the suffering zebrafish organism responds by bypassing the first conversion of essential LA and ALA precursors via the alternate route involving its elongation, followed by Δ8 desaturation instead of Δ6 desaturation followed by elongation. The mechanism of these reactions remains to be elucidated further. We speculate that a Δ6 desaturase activity cell deficient in Δ6 desaturase activity has a lower risk of losing GLA, when it might not be met with an appropriate elongase. Firstly, LA becomes subject to elongation in excess to obtain a higher probability for this molecule to meet with its FADS2 desaturase (performing desaturation in the Δ8 position, in this case). Even if the *fads2*-defective organism suffers from an inadequate GLA abundance, bypassing the first LA (and ALA) conversion, this seems to be an effort to rescue its conversion to a DGLA (and SA) molecule, which is an important substrate for not only other downstream omega-6 (and omega-3) LC-PUFAs, but also eicosanoids. On the other hand, the primary aim of the present study was to analyze zebrafish with an impaired function of the FADS2 enzyme, which supplies the organism by GLA by converting the essential LA. The content of GLA, unlike other LC-PUFAs of the n-6 series, is very low in food, hence the proper function of the FADS2 enzyme is crucial for the maintenance of the health of a vertebrate organism. Considering the fact that GLA serves precursors of prostaglandins and leukotrienes, we intended to answer the question of whether the endogenous LC-PUFA biosynthesis is indispensable in addition to the standard food conditions in which ω6 LC-PUFA biomolecules are normally delivered.

Recently, a study was conducted, which confirmed that salmon (*Salmo salar*) Δ6 FADS2 possesses Δ8 desaturation activities towards its substrates C20:2 d11, 14 and C20:3 d11, 14, 17 in vivo. In agreement with our observation, both substrates accumulate in *fads2* salmon knockouts [40]. Salmon has four *fads2* genes encoded in the genome Δ6fads2-a, Δ6fads2-b, Δ6fads2-c, Δ5fads2. The authors constructed two CRISPR-mediated partial knockouts *Δ6fads2*-abc/5 with mutations present in all four desaturases and knockout Δ6*fads2*-ab. Both knockouts displayed the degree of gene editing of 50–100%, which means that up to 50% of their desaturases might still work and/or compensate to offset each other. A genetic background that is too complex can enable the study of the milestone at which the organism can choose and divert to alternative routes to sustain a health-endangering situation. These authors suggested that the Δ8 desaturation pathway may be activated by high dietary levels of LA and ALA, and functions together with the Δ6 pathway to enhance the conversion of C18 precursors to downstream LC-PUFAs under limiting conditions. Our data indicate this scenario, since zebrafish partial *fads2* knockouts were able to produce AA, EPA, and DHA, generally considered as the most important products of LA and ALA conversions in LC-PUFA biosynthesis. Contrary to our expectations, AA, EPA, and DHA did not dramatically decrease. This could mean that the zebrafish body was capable of taking and store these biomolecules from the standard diet provided, and/or at the same time produce endogenously by the remaining LC-PUFA biosynthesis, something we did not intended to discern. The main conclusion of our observation is that not solely the abundance of AA, EPA, and DHA, or their relative ratios, counted towards the essential substrates AA/LA and EPA/ALA or DHA/EPA and the balance between ω6 and ω3 in fish, and, especially in brood stock, this should be considered in regard to state of health. Our study emphasizes the importance of the first step of LC-PUFA C18 substrates and suggests that the diversion through the Δ8 alternative pathway might at least be a sign of reproductive success impairments. Our study highlights partial *fads2* knockout zebrafish as a uniquely suited model organism, from which the investigation into disease pathogenesis, genetic diseases, and disorders caused by FADS2 function insufficiencies due to mutated *fads2* genes might considerably advance.

## Figures and Tables

**Figure 1 genes-13-00700-f001:**
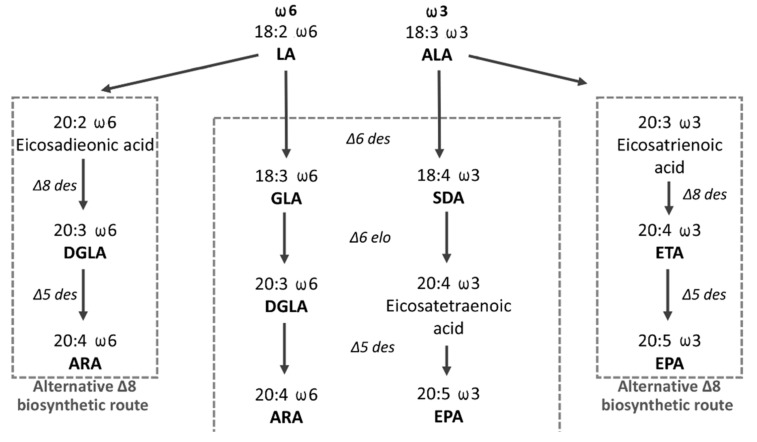
The scheme of the LC-PUFA endogenous biosynthesis of the ω6 and ω3 series from dietary C18:2 Δ9,12 (linoleic acid, LA) and C18:3 Δ9,12,15 (ALA) fatty acids. Dihomo γ-linolenic acid (DGLA): arachidonic acid (ARA): γ-linoleic acid (GLA): stearidonic acid (SDA): eicosapentaenoic acid (EPA): eicosatrienoic acid (ETA).

**Figure 2 genes-13-00700-f002:**
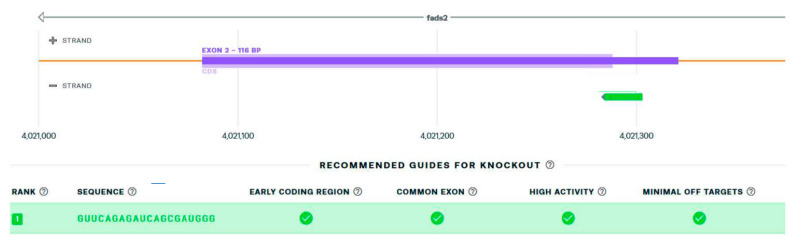
The best hit of single guide RNA for targeting the ATG start codon of the *fads2* zebrafish gene (green box by Synthego).

**Figure 3 genes-13-00700-f003:**
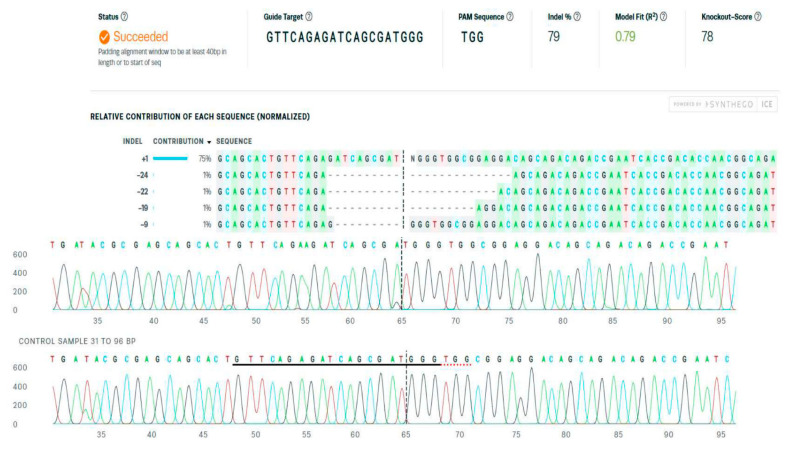
The Inference of CRISPR Edits (ICE) software output of the analyses of the Sanger sequencing data on the *fads2* gene part flanking the mutated ATG start codon in fish representative (here, crispant female No. 1) (ice.synthego.com, accessed on 1 April 2022).

**Figure 4 genes-13-00700-f004:**
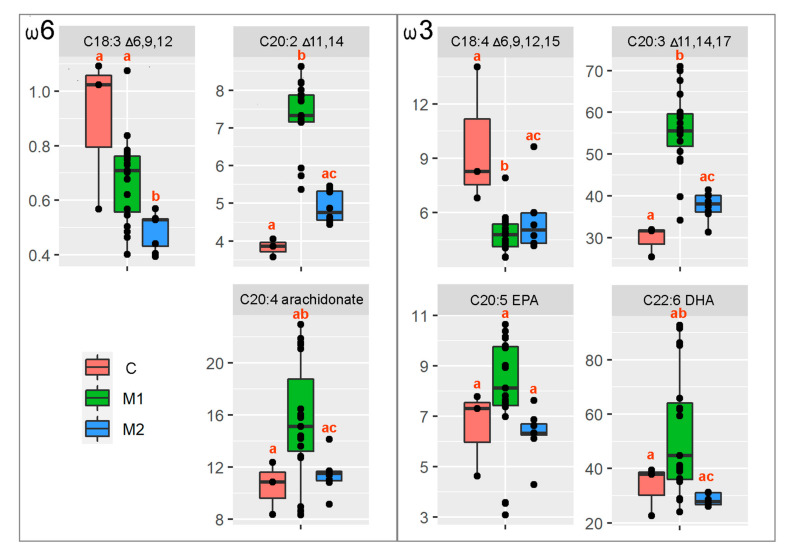
Box and whisker plots for the given LC-PUFA ratio summarizing the median value (black solid horizontal line), first and third quartiles, and minimum and maximum values. The different letters above particular box and whisker plots represent the significant statistical difference among the particular groups.

**Figure 5 genes-13-00700-f005:**
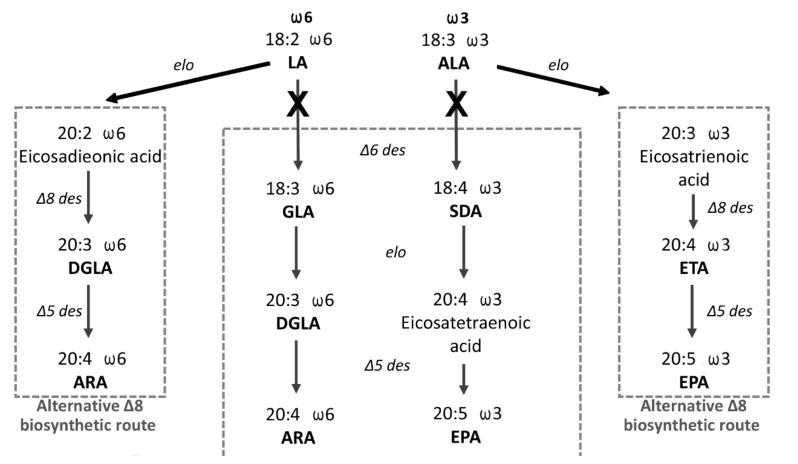
Scheme of the diversion of the LC-PUFA biosynthetic initial step of conversion of LA and ALA from the Δ6 pathway, rather than the Δ8 pathway in fads2 partial zebrafish mutants.

**Figure 6 genes-13-00700-f006:**
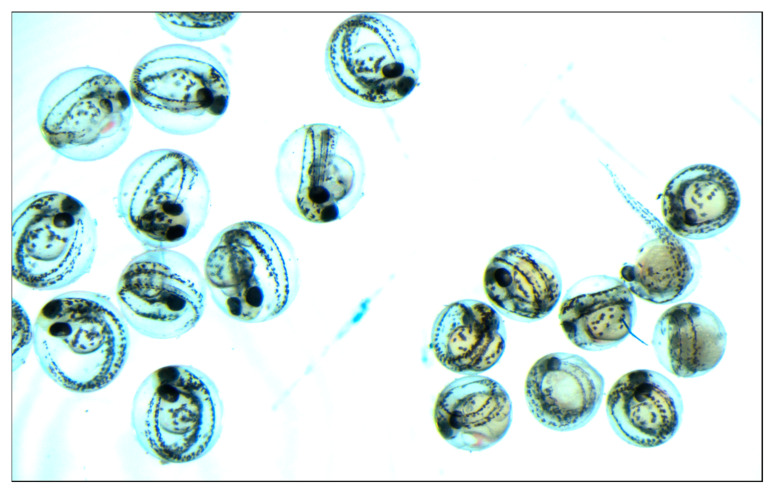
Embryonic lethality and affected development in embryos from crispant female x WT male inter-crosses (right side of the picture) and WT x WT (left side of the picture).

**Table 1 genes-13-00700-t001:** The statistical results of the Kruskal–Wallis tests (*H*-value) and one-way ANOVA (*F*-value) for particular PUFAs.

	*H*-Value (*F*-Value)	Number of d.f.	*p*-Value
C18:3 Δ6,9,132	9.99	2	0.007
C20:2 Δ11,14	20.39	2	<0.001
C20:4, arachnidonate	9.37	2	0.009
C18:4 Δ6,9,12,15	7.56	2	0.023
C20:3 Δ11,14,17	17.50	2	<0.001
C20:5, EPA	2.12	2.27	0.139
C22:6, DHA	11.36	2	0.003

## Data Availability

Not applicable.

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
