# Peer review of "Partial fads2 Gene Knockout Diverts LC-PUFA Biosynthesis via an Alternative Δ8 Pathway with an Impact on the Reproduction of Female Zebrafish (Danio rerio)"

_genes, 2022, doi:10.3390/genes13040700_

Round 1

Reviewer 1 Report

I added all my comments directly to the manuscript file (attached).

Author Response

Dear reviewer 1,

We are very grateful to you for reviewing our manuscript and we greatly appreciate your knowledge of the topic. We have focused to improve our manuscript according to your valuable comments and suggestions.

Comments and suggestions:

L55

Please remove hyphen between ω and 6, ω and 3.

- Removed.

L56

Please use superscripts to indicate the position of double bonds with Δ instead of „d“. „d“ would possibly be confused with a deuterium labelling. Or, I´m just wondering why the authors did not simply use ω6 or ω3 instead of numbering all the double bonds. For example, 18:3ω6 for 18:3Δ6,9,12 is absolutely fine to use since those „typical“ PUFA are normally „methylene-interrupted fatty acids.

- We have used Δ instead of „d“ accordingly. We originally started numbering double bonds consistently with GC data outputs.

L60

The Elovl5 (or Elovl2) enzymes isolated from many fish species have shown to possess elongation activity towards LA and ALA. This would be a general trait of Elovl5 enzyme in fish. If you please you can see a couple of paper below.

               - We are very grateful for this comment, went through recommended publications and added the information to our text with references to the literature cited.

L111

This is not true. If you please can check the paper below. Lopes-Marques, M et al. Retention of fatty acyl desaturase 1 (fads1) in Elopomorpha and Cyclostomata provides novel insights into the evolution of long-chain polyunsaturated fatty acid biosynthesis in vertebrates. Bmc Evol Biol 18, 157 (2018).

- Our statement in the introductory sentence of the paragraph was, of course, our mistake. We have changed the sentence. Thus, instead of „ All the fish fatty acids desaturase (fads2) genes cloned to date are homologous to mammalian fads2…“ we newly state „ Substrate specificities of FADS2 differ among fish species…“ and do not mention the fact, that fads1 gene has been confirmed as not being lost during the evolution in some species. Given the fact that we addressed that evolutionary acpect in our previous review on Fads2 (doi:10.3390/biom10020206) we are very grateful that You pointed out this mistake to us.

L117

Why? This pathway has been suggested in many fish species as their Fads2 and Elovl5 activities. Please see my previous comment on Elovl5 activity towards LA and ALA.

- We have removed the part of the sentence“ …, the later one not being elucidated completely so far.“, not to be incorrect.

L118

Zebrafish would rather be „as suitable model to study impacts of fads gene function in general (not specifically fads2)“ since they do not „naturally“ possess fads1. This means fads2 would be a sole enzyme with desaturation activity towards PUFA in zebrafish.

- We completely agree with that. We acknowledge that our wording is misleading and we have reformulated the text.

L207

I do not understand why this customer service lab needed to use GC x GC/MS to analyse FAME. The FAME analysis can sufficiently be achieved by a normal conventional GC with flame ionisation detector. MS is sometimes required to characterise the structure of unusual FA but normally, compering retention times with commercial standards would be enough to identify usual FA. Please elaborate more on the method such as GC temperature/gas pressure condition, GC columns, MS settings. Plus, it would be difficult to identify LC-PUFA methyl ester by EI MS due to too much fragmentation occured.

- It is true that FAMEs could have been analysed using GC-FID, however, the service lab at Charles university decided to perform the GCxGC-MS experiment because of several reasons: 1) Besides of targeting of the analysis to a limited set of FAMEs, we were interested in looking for all detectable FAMEs. This data is not used in this study. The GCxGC experiment provides two-dimensonal visualization of the separation. This type of the two-dimensional separation produces a structured chromatogram, that together with EI mass spectrum enables estimation of the structure of unknown FAMEs, those that do not match with a commercial mix of FAME standards used in this experiment (GLC 744, NU-CHEK-PREP). 2) GCxGC experiment achieve a higher sensitivity due to a separation a background contamination, particularly a column bleeding. 3) The used instrumentation (Pegasus 4D, Leco) is equipped with MS detection only. FAMEs studied in this work were identified using a combination of matching a two-dimensional retention behavior with commercial FAME standards (GLC 744, NU-CHEK-PREP) and mass spectra.

L260

The authors need to analyse the expression level of elovl5 gene to confirm Δ8 pathway was bona fide activated due to the lack of initial Δ6 desaturation. Unfortunately, no direct evidence that the zebrafish Elovl5 possesses elongation activity towards LA and ALA, but nevertheless, many fish Elovl5 showed those activities as I mentioned in my previous comment. The FA data indeed demonstrated increase of 20:2n-6 and 20:3n-3 in crispants, which are direct elongation products from 18:2n-6 and 18:3n-3, respectively.

- After considering your comments, we are now aware that in the very design of the experiment, we were too focused only on desaturase enzymes. Elongases, although an integral part of the entire biosynthetic pathway, have been neglected by us. We agree with you, but unfortunately, we do not have RNA isolated from eggs produced by crispant and wt females to intend to proceed to optimize RT-qPCR or ddPCR to establish expression levels of elovl5 gene. Moreover, the experiment on crispants run in spring 2020, slowed down due to covid pandemic situation and finally stopped due to the age of fish, which do not live any more. At this moment, we are unable to meet the requirement to measure elovl5 gene expression in the near future. That would mean to create new zebrafish crispants, race them until adulthood and mate them to get eggs. We have thought about it already, that our work would benefit from transcriptomic data as we have mentioned in the discussion (L327). Taking these facts into consideration, we would like to satisfy you at least in part, by reformulating our conclusions in Results (L337) as follows:

Instead of “zebrafish partial fads2 mutants preferentially convert LA and ALA in an Δ8-pathway…” we say, what our data actually say: “ Our data demonstrated increase of 20:2Δ11,14 and decrease of GLA in ω6 pathway and increase of 20:3Δ11,14,17 and decrease of SDA in ω3 pathway.”

L337

I cannot agree on these sentences because the situation in zebrafish and most of marine fish (more precisely, most of Acanthopterygian marine fish) are completely different in terms of Fads2 regioselectivity. Fads2 from most of marine fish generally possess only Δ6 (and Δ8) desaturase activity so Δ5 pathway is naturally absent in those fish species, and hence AA and EPA are anyway impossible to be synthesized from LA and ALA, respectively. As zebrafish Fads2 possessing Δ6Δ5 (and Δ8) multifunctional activity, the crispant established in the present study still is capable of catalysing not only Δ6 pathway but also Δ5 pathway to achieve AA and EPA biosynthesis from LA and ALA, respectively.

- Thank you again for a valuable comment and helping us to write the manuscript better. We have omitted misleading and erroneous sentences from the paragraph not to compare marine fish and zebrafish in their activity of FADS2.

L361

But Δ8 desaturase activity are anyway still present in some cells in the crispants established in the present study because of using partial fads2 knockout.

- Accordingly, we have added the word partly to the end of the sentence to reflect that fact.

Yours sincerely,

Zuzana Bláhová

Reviewer 2 Report

The MS was on the effect of partial knockout of fads2 on the egg fatty acid composition and the reproduction of zebrafish. There are some question on the MS:

1: L266 Authors states that "AA, EPA and 266 DHA are abundant in excess in the artificial diet given ad libitum to fish", therefore, the fatty acid composition of the diet needs to be added

2: Since the research is related with the effect of fads2 knock out on the reproduction of zebrafish, is there any information on the difference of reproduction parameters, for instance, the total eggs produced per female per gram, survival rate of larvae?

Minors

Please check again the font of the MS, such as the blank space between "fish" and "[24]"(L114), the extra space at the beginning of L179 and the alignment of the reference from 23 (L488)

Author Response

Dear reviewer 2,

We are very grateful for reviewing our manuscript. We have focused to improve our manuscript according to your comments and suggestions.

Comments and suggestions:

1: L266 Authors state that “AA, EPA and DHA are abundant in excess in the artificial diet given ad libitum to fish”, therefore, the fatty acid composition of the diet needs to be added.

At the beginning of our experiment, we have analysed the fatty acid composition of our zebrafish diet, however, we hesitate to publish the data without having a manufacturer´s agreement.  We added the information on the feed manufacture including the link http://www.zebrafish.skrettingusa.com in the manuscript text.  Skretting´s GEMMA Micro is a patented diet offering complete nutrition to facilitate early weaning of zebrafish larvae to adult fish. Complete integration of the GEMMA Micro program results, according to the producer web site, in healthy, robust, and highly fecund zebrafish. Fatty acid composition can be found for example on https://mbki.com/skretting-the-complete-nutrition-for-all-lifestages-of-zebrafish/. Thank you for your comment, we filled in the missing information.

2:  Since the research is related with the effect of fads2 knock out on the reproduction of zebrafish, is there any information on the difference of reproduction parameters, for instance, the total eggs produced per female per gram, survival rate of larvae?

Total eggs produced per crispant female with knock out score 80% was not possible to count properly because of the severely compromised integrity of the membrane. Many of those eggs were falling apart, even indistinguishable. Another reason, why we finally do not have publishable data on total eggs produced by female is, that these “high knock out score females” need to be handled extremely carefully during stripping their bally by hand. Not all of the eggs were stripped for sure, hence, the total egg amount was questionable. Anyway, challenging is the phenomenon of masoacism itself. Not all of the eggs have mutated genome and if, theoretically each could exhibit different percentage of indels and mutagenesis efficiency. Indeed, counting the survival rate of larvae produced by crispant female would need to determine in each of the embryo/lecithotrophic larvae, if it is fads2 mutant by evaluation of chromatograms from Sanger sequencing of total DNA extracted from “freshly” died embryo/larvae. To sum up, according to our experiences, the potential to get valuable reproduction parameters will be higher first in confirmed biallelic mutants - progenies of crispant x crispant inter-crosses, ones we succeed to produce them.

Minors

Please check again the font of the MS, such as the blank space between “fish” and “[24] (L114), the extra space at the beginning of L179 and the alignment of the reference from 23 (L488).

  • We have changed accordingly.

Yours sincerely,

Zuzana Bláhová

Round 2

Reviewer 1 Report

I am satisfied by the author's reply to my comments.